# Point-of-Care Ultrasound in Airway Evaluation and Management: A Comprehensive Review

**DOI:** 10.3390/diagnostics13091541

**Published:** 2023-04-25

**Authors:** Judy Lin, Ryan Bellinger, Andrew Shedd, Jon Wolfshohl, Jennifer Walker, Jack Healy, Jimmy Taylor, Kevin Chao, Yi-Hsuan Yen, Ching-Fang Tiffany Tzeng, Eric H. Chou

**Affiliations:** 1Department of Emergency Medicine, Baylor Scott & White All Saints Medical Center, Fort Worth, TX 76104, USA; judy.lin@bswhealth.org (J.L.); ryan.bellinger@bswhealth.org (R.B.); andrew.shedd@bswhealth.org (A.S.); jwalker@ies.healthcare (J.W.); yihsuan.yen@bswhealth.org (Y.-H.Y.); tiffany.tzeng@bswhealth.org (C.-F.T.T.); 2Burnett School of Medicine, Texas Christian University, Fort Worth, TX 76109, USA; jack.healy@tcu.edu (J.H.); j.m.taylor@tcu.edu (J.T.); kevin.chao@tcu.edu (K.C.); 3Department of Emergency Medicine, Baylor University Medical Center, Dallas, TX 75246, USA

**Keywords:** ultrasound, point-of-care, difficult airway, endotracheal intubation, cricothyrotomy

## Abstract

Airway management is a common and critical procedure in acute settings, such as the Emergency Department (ED) or Intensive Care Unit (ICU) of hospitals. Many of the traditional physical examination methods have limitations in airway assessment. Point-of-care ultrasound (POCUS) has emerged as a promising tool for airway management due to its familiarity, accessibility, safety, and non-invasive nature. It can assist physicians in identifying relevant anatomy of the upper airway with objective measurements of airway parameters, and it can guide airway interventions with dynamic real-time images. To date, ultrasound has been considered highly accurate for assessment of the difficult airway, confirmation of proper endotracheal intubation, prediction of post-extubation laryngeal edema, and preparation for cricothyrotomy by identifying the cricothyroid membrane. This review aims to provide a comprehensive overview of the key evidence on the use of ultrasound in airway management. Databases including PubMed and Embase were systematically searched. A search strategy using a combination of the term “ultrasound” combined with several search terms, i.e., “probe”, “anatomy”, “difficult airway”, “endotracheal intubation”, “laryngeal edema”, and “cricothyrotomy” was performed. In conclusion, POCUS is a valuable tool with multiple applications ranging from pre- and post-intubation management. Clinicians should consider using POCUS in conjunction with traditional exam techniques to manage the airway more efficiently in the acute setting.

## 1. Introduction

Airway management is a critical and essential component of practicing emergency medicine [1]. It involves a technically difficult skill used often in sub-optimal circumstances, with the goal of ensuring adequate ventilation and oxygenation in critically ill patients. Each step must be performed efficiently and accurately to ensure the best possible outcomes. Because of these challenges, emergency airway management can be a high-risk procedure and may lead to severe morbidity and mortality in difficult intubations [2]. The incidence of complications and adverse events during airway management is significant, with up to 12% of intubations resulting in complications in emergency departments (ED) in the United States [3].

Point-of-care ultrasound (POCUS) has gained popularity in recent years and has been widely used in EDs as a diagnostic tool and imaging guide for various procedures [4]. Because of its portability and high diagnostic accuracy in a wide variety of applications, Emergency Medicine (EM) physicians consider POCUS to be a crucial component of resuscitation [5]. Ultrasound has also emerged as a promising tool for airway assessment and management due to its familiarity, accessibility, safety, and non-invasive nature. It can assist physicians in identifying relevant anatomy with objective measurements of airway parameters and guiding airway interventions with dynamic real-time images [6,7].

Examples of ultrasonographic applications of airway management are multifold. Ultrasound can be used to assess the airway prior to intubation or procedural sedation, helping identify patients who may have an otherwise unanticipated difficult airway. Further, it can be used to confirm proper placement of an endotracheal tube (ETT), which is especially important in situations where confirmation with end-tidal capnography may be unreliable, such as cardiac arrest [8]. Additionally, ultrasound can be used to identify ETT misplacement, such as esophageal intubation or mainstem intubation [9,10]. Lastly, ultrasound can also be used to identify the cricothyroid membrane (CTM), a crucial step in preparing for a “cannot intubate, cannot ventilate” scenario [11]. By using ultrasound to locate the CTM, physicians can perform a cricothyrotomy quickly and accurately in the event that intubation is not possible.

In this review article, we aim to provide a comprehensive overview of the current evidence on the use of ultrasound in airway management and to identify areas for future research. We will discuss the current literature on the use of airway ultrasound in emergency medicine and its potential benefits and limitations in clinical practice.

## 2. Methods

A PubMed and EMBASE literature research was conducted for English language articles published from 1 January 1996 to 20 March 2023, using the keyword “ultrasound” combined with several search terms, i.e., “probe”, “anatomy”, “difficult airway”, “endotracheal intubation”, “laryngeal edema”, and “cricothyrotomy”. The most recent systematic reviews, meta-analyses, randomized controlled trials, guideline statements, and studies with larger sample sizes were prioritized for inclusion. EM physicians with more than 5 years of experience in POCUS and critical appraisals of the literature reviewed the articles, and they determined which studies to include for the review by consensus, with a focus on EM relevant articles.

## 3. Probe Selection and Technique

The two most commonly used ultrasound probes for visualization of the upper airway include the linear and curvilinear probe. Identification of superficial structures such as the CTM, vocal cords, and the epiglottis are better visualized using a standard 5–14 MHz high-frequency linear probe. While higher frequencies allow for better resolution of superficial structures, they have poorer resolution of deeper structures, such as the tongue base [12]. The standard 4–10 MHz curvilinear probe is better suited to evaluate the tongue base due to the ability of lower frequency soundwaves to penetrate into deeper tissues [12]. In addition, certain airway measurements such as the hyomental distance (HMD) should be performed with the curvilinear probe, as the longer length of the probe footprint can easily visualize both the hyoid and mentum in one image for accurate distance measurement. 

The ultrasound probe can be placed in the transverse, sagittal, or parasagittal position. The upper airway can be evaluated in both transverse and sagittal axes from the suprasternal notch to the mentum. The parasagittal position may be helpful to aid in the performance of a midline procedure, such as a cricothyroidotomy. The patient should be positioned supine, and the patient’s neck can be in a neutral, ramped, or hyperextended position. Having the patient ramped with their head extended tends to increase the surface area available to manipulate the probe, allowing for improved ease of imaging. However, if the patient is in respiratory distress, a semi-recumbent or seated position can also be used. Ample gel should be used to decrease the pressure of the probe applied to the neck and to prevent the occurrence of air pockets between the protuberances of the tracheal rings and thyroid cartilage. Color Doppler can be used to identify important vascular structures. Patient maneuvers can be used to better elucidate anatomy. For example, having the patient swallow may aid in the visualization of the esophagus and having the patient phonate can help to evaluate vocal cord function.

## 4. Upper Airway Sono-Anatomy

Upper airway ultrasound allows for the identification of important upper airway structures such as the trachea, esophagus, tracheal rings, cricoid cartilage, CTM, thyroid cartilage, vocal cords, hyoid bone, epiglottis, and tongue [13]. In addition, ultrasound measurements of the upper airway have been shown to have both high inter- and intra-operator reliability as well as accuracy when compared to cadaver models [14].

The following five views can be used to assess the upper airway: suprahyoid, thyrohyoid, thyroid, cricothyroid, and suprasternal (Figure 1). Depending on the specific reason for upper airway evaluation, fewer views can be selected to answer a focused question. 

### 4.1. Suprahyoid View

The suprahyoid view is located above the hyoid, from the hyoid to the mentum, and is used to measure hyomental distance (HMD), distance to tongue, and tongue thickness (Figure 2). The measurements on this view may vary with changes in patient positioning [15]. The main structures seen on this view include the hyoid, mentum, tongue, mylohyoid muscle, and geniohyoid muscle.

On sagittal view, with the indicator towards the mentum, the hyoid is seen on the right of the image as a hyperechoic protuberance with posterior shadowing. To the left of the image is the symphysis menti, which also appears as hyperechoic with posterior shadowing. The hypoechoic tongue lies in between, just above the palate. Superficial to the tongue are the hypoechoic mylohyoid and geniohyoid muscles [12]. 

### 4.2. Thyrohyoid View

The thyrohyoid view is located through the thyrohyoid membrane and is used to visualize the epiglottis (Figure 3). The main structures seen on this view include the strap muscles, thyrohyoid membrane, pre-epiglottic space (PES), and the epiglottis. 

At this level, the hypoechoic strap muscles converge medially and bridge over the thyrohyoid membrane, which lies in between the strap muscles and the PES. The PES is a hyperechoic collection of adipose tissue immediately deep to the strap muscles and superficial to the epiglottis. The epiglottis appears as a thin hypoechoic stripe lying just anterior to a hyperechoic air-mucosa (A-M) interface. The entire image appears as a “small face sign” where the strap muscles appear as eyes and the epiglottis appears as the mouth [16].

### 4.3. Thyroid View

The thyroid view is located over the thyroid cartilage and is used to visualize the vocal cords (Figure 4). The main structures seen on this view include the vocal cords, arytenoid cartilage, and strap muscles. As the thyroid cartilage becomes more calcified with age and shadowing from calcifications may obstruct this view, visualization of the vocal cords can also be performed through the cricothyroid membrane and angling the probe skilfully [12].

The strap muscles are seen superficially to the vocal cords and have a hypoechoic striated appearance. In addition, the vocal cords and arytenoid cartilage are both identifiable at the level of the thyroid cartilage. The thyroid cartilage appears just below the strap muscles and forms a triangular shape. Deep to the thyroid cartilage are the vocal cords, which form an isosceles triangle with central shadowing. The arytenoid cartilages are deep to the vocal cords and appear hyperechoic with shadowing [12].

### 4.4. Cricothyroid View

The cricothyroid view is located between the cricoid cartilage and the thyroid cartilage (Figure 5). The main clinical utility of this view is to locate the CTM and any structures overlying the CTM for cricothyrotomy. The main structures seen from this view include the cricoid cartilage, CTM, and the thyroid cartilage.

In the transverse plane, the “thyroid-airline-cricoid-airline” (TACA) protocol can be used to identify the CTM. First, the prominent, triangular thyroid cartilage is visualized. It may be partially or fully calcified depending on age and therefore has posterior shadowing. Moving caudally, the hyperechoic linear CTM can be found. More caudally, the hypoechoic lateral edges of the cricoid cartilage will appear to converge anterior to the CTM. Scanning cranially, the CTM can then be identified again [17]. In the sagittal plane, the thyroid cartilage will plunge from superficial to deep as it moves inferiorly until its border with the CTM. The cricoid cartilage will appear inferior to the CTM as a hypoechoic ovoid structure, and the tracheal rings will appear as smaller evenly spaced intermittent hypodensities inferior to the cricoid cartilage. This appearance in sagittal view has been repeatedly described as “beads on a string” [17].

### 4.5. Suprasternal View

The suprasternal view is located just above the suprasternal notch and is the best location to visualize ETT location and placement, and to measure the tracheal diameter (Figure 6). The main structures seen on this view include the trachea, esophagus, thyroid, carotid artery and internal jugular veins, vertebra, and sternocleidomastoid muscles.

On the transverse view, the trachea appears as a curvilinear convexity with a hyperechoic air–mucosa (A–M) interface and posterior reverberation artifact. The lobes of the thyroid can be seen immediately lateral to the trachea with the thyroid isthmus bridging over the trachea anteriorly. Posterior to the lateral thyroid lobes are the common carotid arteries and internal jugular veins, with the sternocleidomastoid muscle lying superficial to the vessels. Color Doppler can be used to confirm flow within the vascular structures. The esophagus is usually located in the posterior and to the anatomical left of the trachea appearing with both hyper and hypoechoic components [18]. Visualization of both trachea and esophagus at this level is important as the presence of a “double tract sign” with two hyperechoic (A-M) interfaces indicate the improper placement of the ETT into the esophagus during intubation [9,19].

## 5. Sonographic Assessment of the Difficult Airway

Airway management and intubation are critical skills among several specialties of medicine, especially so in EM, anesthesia, and critical care. The incidence of failed intubation remains high, around 1 in 50 to 100 patients in the ED, intensive care unit (ICU), and prehospital setting [20]. The difficult airway is multifactorial, difficult to predict, and is associated with high morbidity and mortality [20]. A standardized methodology is generally used to assess patients for a difficult airway, and includes the evaluation of aspiration risk, demographics, comorbidities, and a physical assessment of facial and jaw landmarks, Mallampati score, neck mobility, and any abnormal anatomy [21,22]. However, despite this well described methodology for assessing the likelihood of a difficult airway, some studies found that over 90% of difficult airways are unanticipated [23].

Clinical screening tests and measurements of airway anatomy used for difficult airway evaluation have low sensitivity and specificity [15,16,24,25,26]. This may be in part due to the difficulty in identifying landmarks, especially in the obese patient population. Sonographic identification of upper airway anatomy has been shown to outperform that of palpation and has good correlation with computed tomography measurements [6,15,27].

Grading the difficulty of intubation can be subjective, and, therefore, a common outcome measure used to assess difficult airways is the Cormack-Lehane (CL) grade. The CL grade classifies the degree of vocal cord visualization on a scale from 1 to 4, with grades 3 and 4 defined as difficult laryngoscopy [25]. In recent years, several studies have investigated the ability of ultrasound to predict difficult laryngoscopy as a component of preoperative evaluation (Table 1). Ultrasound measurements within three domains have been identified to correlate with the CL grade and include the anterior neck soft tissue thickness domain (TTD), anatomic position domain (APD), and oral space domain (OSD) [26]. Measurements of the TTD evaluate the submandibular space and therefore assess the curvature that the laryngoscope is required to traverse across the tongue. Measurements of APD are dynamic and vary with patient positioning in neutral, ramped, and hyperextended positions. They reflect the degree to which the tongue can be displaced and the neck extended in order to optimize the view of the glottis. Measurements of the OSD evaluate the size of the oral cavity and the tongue, which may contribute to difficult laryngoscopy [26]. Several meta-analyses have found that ultrasound measurements of these three domains are significantly different in those with and without difficult laryngoscopy (DL) [26,28,29,30].

Ultrasound measurements of the TTD that have been studied include the distance from skin to epiglottis (DSE), skin to anterior commissure of the vocal cords, skin to vocal cords, skin to thyroid cartilage, and skin to thyrohyoid membrane [16,26,31,34,54,55,56,57,58]. The APD measurements include the HMD, HMD ratio (HMDR), condylar translation, and the angle between the glottis and epiglottis [27,44,46,55,57,59,60]. The HMD is found by placing the curvilinear probe on mid-sagittal axis in the suprahyoid position so that both the hyoid bone and mentum can be seen. The distance between the posterior border of the symphysis menti and the anterior border of the hyoid bone is the HMD (Figure 2B) [27]. The HMD ratio (HMDR) can be found as HMDR1, which is the ratio of HMD in ramped position to the HMD in neutral, or as HMDR2, which is the ratio of HMD in maximal head extension to the HMD in neutral position. Measurements of the OSD include tongue thickness, distance from skin to tongue, and tongue volume. Out of all domains, Bhargava et al., found that the dynamic measurements of APD (Sensitivity 74%, Specificity 86%) had the best pooled sensitivity and specificity compared to that of TTD (Sensitivity 76%, Specificity 77%) and oral space measurements (Sensitivity 53%, Specificity 77%) [26].

Amongst all TTD measurements, DSE has the most data supporting its prediction for difficult laryngoscopy and has been studied across many different ethnicities and patient populations [15,26,42]. DSE is evaluated in the thyrohyoid view with the probe in transverse orientation and reflects the thickness of the pre-epiglottic space (Figure 3b) [16]. Fernandez-Vaquero et al. found that DSE in the sniffing position ≥ 2.48 cm was the best preoperative predictor of a CL grade ≥ 2b with a sensitivity of 91.3% and specificity of 96.9% [15]. Ni et al. evaluated DSE compared to the distance between skin and thyroid cartilage, and distance between thyroid cartilage and epiglottis, and found that a DSE of ≥2.36 cm was the best predictor of a difficult CL grade with a sensitivity of 81.8% and specificity of 85.6% [42]. Falcetta et al. evaluated DSE in the neutral position compared to the distance from skin to thyrohyoid membrane and from skin to vocal cords, and found that a DSE of ≥2.54 cm was the best predictor of a CL grade ≥ 2b with a sensitivity of 82% and a specificity of 91% [42]. A meta-analysis by Carsetti et al. found that the pooled sensitivity and specificity of DSE for difficult laryngoscopy was 82% and 79%, respectively [28].

The best predictors of difficult laryngoscopy within the APD measurements include HMD and the HMDR2. More recent meta-analyses have shown HMD to be the single most reliable predictor of difficult laryngoscopy [26] and difficult intubation (DI) [7,61]. Wu et al. found that the accuracy of HMD for predicting a CL grade ≥ 3 was significantly greater than that for other measurements, including DSE [61]. According to their study, an HMD ≤ 5.29 cm had a sensitivity of 96.7% and specificity of 71.6% [61]. In a recent meta-analysis, Gomes et al. found that out of 26 ultrasonographic airway parameters, HMD was the most consistent and reliable parameter for predicting a difficult airway and was significantly associated with difficult laryngoscopy [7]. Petrisor et al. found that HMDR1 ≤ 1.12 was also able to predict difficult laryngoscopy with a sensitivity of 75% and a specificity of 76.2% [60]. Other studies found that HMDR2 ≤ 1.085 (Sensitivity 75%, Specificity 85.3%), and ≤1.23 (Sensitivity 100%, Specificity 90.5%) correlated with difficult laryngoscopy [7,44,60].

In terms of OSD parameters, tongue thickness is likely the best predictor of difficult laryngoscopy compared to tongue volume [7]. Yao et al. found that a tongue thickness > 6.1 cm had a sensitivity of 75% and a specificity of 72% for predicting difficult intubation [51].

With increasingly encouraging data regarding the aforementioned parameters, we summarize a standardized protocol to evaluate for difficult laryngoscopy: *Difficult Airway Evaluation with Sonography (DARES) Protocol* (Figure 7). Our literature search reveals a long list of upper airway parameters without any consistent or concise protocols for airway assessment. The DARES Protocol is a succinct, standardized ultrasound examination to predict difficult laryngoscopy based on upper airway measurements with the most robust data. The protocol involves only two views of the upper airway: the thyrohyoid and the suprahyoid views. Measurements selected for this protocol include the DSE, HMD, HMDR1, HMDR2, and tongue thickness, which cover all three domains of TTD, APD, and OSD [15,26,42]. Future research is needed to investigate the utility of the DARES protocol and its potential to predict the difficult airway in conjunction with other clinical algorithms.

## 6. Confirmation of Endotracheal Intubation

Tracheal intubation serves as definitive airway control in emergent resuscitations and respiratory failure. Confirmation of proper tube placement in the trachea is essential at the time of intubation. One prior study demonstrated that approximately 3.3% of all emergency intubations are esophageal intubation, which can lead to significant morbidity or death [3]. According to the 2016 American College of Emergency Physician (ACEP) policy statement, traditional physical examination methods are not sufficiently reliable to confirm endotracheal tube placement [62]. Waveform capnography is the current standard for confirming endotracheal intubation. However, it can be significantly affected by low cardiac output, low pulmonary blood flow, or epinephrine use [63]. For patients in cardiac arrest or with otherwise markedly decreased perfusion, both continuous and non-waveform capnography can be less accurate or inconclusive. In these situations, other confirmatory methods should be used [62].

Ultrasound has been recognized in previous studies as a promising and highly accurate method for confirmation of the correct endotracheal tube (ETT) placement (Table 2) [64,65]. This is performed by either directly and dynamically scanning the anterior neck during the intubation, indirectly by looking for lung-sliding of the pleura or diaphragmatic movement, or by combining these techniques [9,65,66]. For confirming the location of ETT placement, tracheal ultrasound has demonstrated a sensitivity of 98.7% and specificity of 97.1% among adult patients, and a sensitivity of 92–100% and a specificity of 100% among pediatric patients [64,67]. This assessment showed no significant differences between cardiac arrest settings versus non–cardiac arrest settings [8].

When performing tracheal ultrasound, the position of the trachea can be determined by a hyperechoic A–M interface with a reverberation artifact posteriorly (comet-tail artifact) (Figure 8A). The ETT position can be defined as (1) tracheal, if only one A–M interface with comet tail artifact and posterior shadowing is observed (Figure 8B), or (2) esophageal, if two A–M interfaces with comet-tail artifacts and posterior shadowing are noted (the “double tract” sign) (Figure 8C) [9]. The “double tract” sign is considered the most commonly utilized finding for esophageal intubation, which offers the advantage of being reliable in both static and dynamic assessments [6]. Some studies have suggested using indirect signs, such as bilateral lung sliding or diaphragmatic elevation for intubation confirmation [66,90]. Recent studies have demonstrated ultrasound to be more accurate than auscultation to assess for mainstem intubation, and more rapid and feasible than chest radiograph [10,91].

Tracheal ultrasound has been included in the recent American Heart Association (AHA) Guidelines as a method to assess ETT placement after intubation [92,93]. Tracheal ultrasound has multiple advantages for airway assessment in emergent settings. First, ultrasound has excellent accuracy for esophageal intubation detection. It can be used when the results of capnography are equivocal, and it may therefore reduce unnecessary intubation attempts in critically ill patients. Second, tracheal ultrasound can be performed in real time as the tube is advanced into the trachea or esophagus. Esophageal placement can thus be identified before any artificial ventilation begins. Third, tracheal ultrasound can be performed during cardiopulmonary resuscitation (CPR) without interruption of chest compressions [74]. To date, tracheal ultrasound has been used widely in acute care settings, including adult and pediatric EDs, ICUs, and the prehospital setting [6,67,71,94].

Nonetheless, there are some limitations to tracheal ultrasound. First, ultrasound scanning is highly operator dependent and may limit the application of the examination due to lack of training. Gottlieb et al. demonstrated that ultrasound can be accurate for airway assessment in thin cadavers but appears less accurate in obese cadavers among ultrasound novices [95]. Prior studies showed that emergency medicine resident physicians who received a 1-h standard airway ultrasound lecture and 8-h of hands-on practice can successfully use ultrasound for airway assessment [9,74]. Thus, we believe a specific airway ultrasound lecture with hands-on practice is required before applying airway ultrasound in clinical practice. Second, the midline approach on the anterior neck may miss the second hyperechoic A–M interface (and thus be interpreted incorrectly as the correct endotracheal tube placement) in cases where the esophagus is located directly posterior to the trachea, a situation noted in as many as 16% of patients [95,96]. Third, although tracheal ultrasound can achieve high sensitivity and specificity to confirm ETT placement in multiple trauma patients, the application can be limited in patients with neck trauma with subcutaneous emphysema, laryngeal injury, or foreign bodies [89]. Prior studies have investigated different ultrasound techniques to improve diagnostic accuracy [96,97,98,99]. To date, there is no statistically significant difference in the diagnostic accuracy between different transducer types, saline versus air cuff inflation, or static versus dynamic scanning techniques [64,97,98,100]. Researchers have recommended several simplified scanning protocols to increase clinical feasibility and accuracy [96,101]. Gottlieb et al. demonstrated that twisting the ETT after the intubation attempt improved tube visualization during static assessment [99]. The diagnostic accuracy, ultrasound scanning time, and operator’s confidence of using the ETT twisting technique on the ultrasound were better than using the traditional static technique [99]. The use of color Doppler to facilitate ETT location identification has been suggested but has not been demonstrated to improve the diagnostic accuracy over the B-mode alone [102].

Based on current guidelines and expert consensus, tracheal ultrasound has been considered one of many confirmation tools for ETT placement assessment [92,93]. While the diagnostic accuracy for tracheal ultrasound is high, it would be most effective in conjunction with other techniques, similar to auscultation and capnography [6].

## 7. Assessment of Laryngeal Edema

Laryngeal mucosal edema occurring after intubation is an important etiology of post-extubation respiratory failure, and is associated with increased mortality, risk of pneumonia, ICU and hospital length of stay, and increased hospital costs [103,104,105,106]. Prior to extubation, a cuff leak test is commonly used to evaluate for laryngeal edema. The cuff leak test is usually performed by deflating the ETT cuff followed by either listening for an audible air leak or looking for a difference between inspired and exhaled volumes of air [107,108,109]. Unfortunately, there is much variability in the studied definition of cuff leak and its accuracy which can lead to variance in extubation strategies and deferral of extubation [107,108,109,110,111,112]. The ultrasound has been studied as a tool to assess laryngeal edema and predict post-extubation stridor [113,114,115].

Some studies suggest that tracheal width measurements may correlate with the development of post-extubation stridor. Shinohara, et al. found that the glottis to the ETT outer diameter ratio < 1 in females performed within three hours prior to or after extubation in 212 patients was associated with airway obstruction symptoms [116]. Ultrasound measurements of tracheal width have been found to correlate highly with CT [113,117].

Ultrasound measurements of laryngeal air column width (ACW) and laryngeal air column width difference (ACWD) have been found to significantly correlate with post-extubation stridor (Figure 9). Ding et al. performed a pilot study in 51 patients, and found that there was significant correlation with both reduced ACW (2 mm versus 0.8 mm) and ACWD (1.3 mm versus 0.5 mm) in the development of post-extubation stridor [118]. Another study of 41 intubated patients compared cuff leak to ACW only and found poor sensitivity and specificity for both cuff leak tests and ACW with a PPV < 20% among both methods [114]. 

In other intubated populations, ultrasound has compared favorably to the cuff leak test in its prediction of post-extubation stridor. In the context of steroid administration for stridor, El-Baradey et al. found good correlation between both ACWD versus cuff leak volume [119]. In post-thyroidectomy patients, ACWD and cuff leak were compared and found to have a high NPV of 99.43% for stridor, suggesting usefulness in this population as well [120]. In a pediatric population of 400 intubated children, a prospective observational study found that an ACWD less than 0.8 mm showed a sensitivity of 93% and specificity of 86% with an accuracy of 91% in predicting post-extubation stridor, as compared to the cuff leak test that had a sensitivity of 61% and specificity of 53% [115]. Finally, a recent meta-analysis found that while there was variation in cutoff values used for ACWD, patients with a smaller ACWD had an increased risk of post-extubation stridor [121].

Similar to the cuff leak test literature, these studies are limited by a smaller number of patients as well as variable cutoff values. Current research in ACWD shows promise in its ability to determine the presence of laryngeal edema as both an individual test as well as a confirmatory test when cuff leak testing is equivocal. 

## 8. Preparation for Cricothyrotomy

Cricothyrotomy is a rare but potentially lifesaving resuscitative maneuver used when endotracheal intubation fails. First pass success rate of cricothyrotomy by anesthesiologists has been reported as low as 36% [6,122]. Inability to identify the CTM contributes substantially to the high failure rate of cricothyrotomy [6]. Though CTM anatomic landmarks are often visible or palpable in thin patients, these landmarks prove difficult to locate in patients with obesity or abnormal neck anatomy. In these difficult cases, ultrasound can help to identify the CTM and guide cricothyrotomy [123].

Ultrasound outperforms visual inspection and digital palpation in the identification of the CTM [124,125,126]. One prior study demonstrated that ultrasound guidance during cricothyrotomy resulted in a five-fold improvement in correct tube placement among subjects with anatomy that was difficult to palpate [127]. A common approach for the ultrasound identification of the CTM is the longitudinal “string of pearls” technique [123,128], in which a linear transducer is placed sagittally in the midline of the neck (Figure 10A). The tracheal rings and cricoid cartilage resemble a “string of pearls.” The cricoid cartilage represents the cranial-most and largest “pearl” in the string. The thyroid cartilage appears just cranial to the cricoid cartilage, and the CTM lies between these two structures. Color flow doppler in this view can identify any overlying vasculature. The string of pearls technique can be augmented with the transverse thyroid-airline-cricoid-airline (TACA) approach as described previously in the sono-anatomy section (Figure 10B–E) [129]. POCUS can also be used to dynamically guide cricothyrotomy when performed with a Seldinger technique [130].

There are several advantages to ultrasound use for cricothyrotomy. Nicholls et al. showed that POCUS trained emergency physicians were able to rapidly (mean time < 30 s) and reliably mark the CTM in obese patients using ultrasound [131]. Ultrasound not only reveals the location but gives the user a reliable depth of the CTM and knowledge of the midline to avoid vasculature if an incision is made [131]. In addition, marking the CTM in extended neck positioning remains a reliable indicator of CTM location after the subject is moved or manipulated, as long as the neck is returned to extension [132,133].

Current evidence suggests that the ultrasound is a quick and reliable means of identifying the CTM, and that skin marking over the CTM remains accurate even after repositioning. We recommend that EM physicians identify and mark the CTM with ultrasound guidance prior to intubation in patients with anatomic predictors of a difficult airway (e.g., obesity, distorted neck anatomy) [6,131]. Doing so will prepare for the worst possible outcome—a “can’t intubate, can’t ventilate” scenario, in which EM physicians must re-position to the front of the neck to perform a lifesaving cricothyrotomy.

## 9. Conclusions

Airway management is a critical and high-risk procedure that may lead to severe morbidity and mortality. Our literature search found POCUS to be an accurate and reliable tool for ETT confirmation, anatomic airway measurements, prediction of the difficult airway, post-extubation laryngeal edema, and cricothyrotomy guidance. We described the DARES protocol, which summarizes the assessment of the upper airway for difficult laryngoscopy. The goal of the protocol is to create a standard, organized evaluation of the upper airway, incorporating the most promising airway parameters. Ultrasound is a safe and useful tool for difficult airway management, and clinicians should also prepare other rescue airway techniques such as bag-valve-mask ventilation or video-laryngoscopy. The hope is that this protocol will prompt future study into the role of POCUS in difficult airway prediction and management. 

## Figures and Tables

**Figure 1 diagnostics-13-01541-f001:**
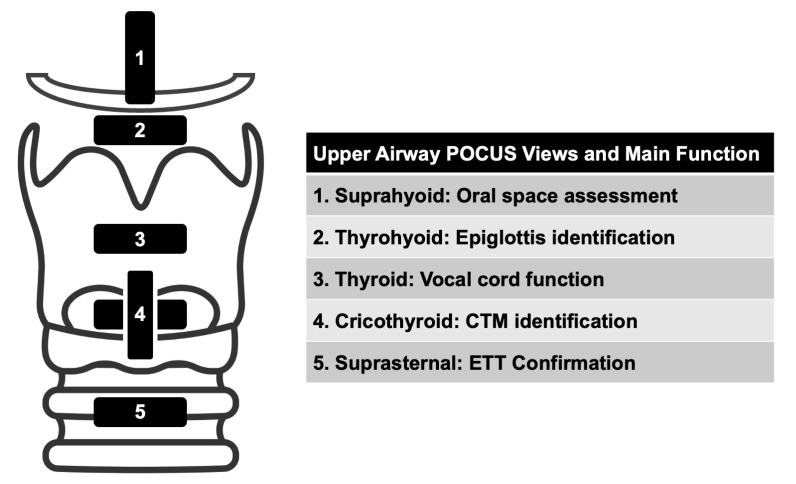
Views to Assess the Upper Airway.

**Figure 2 diagnostics-13-01541-f002:**
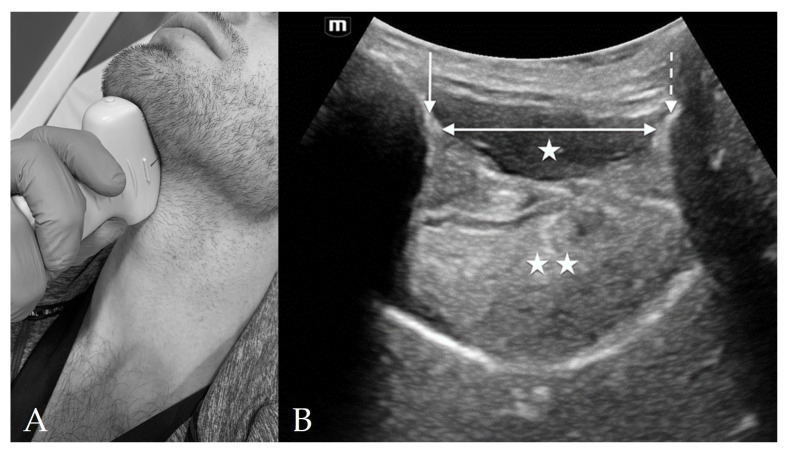
Suprahyoid View. (**A**) Suprahyoid probe placement on the subject’s neck in sagittal orientation. (**B**) Suprahyoid view of anterior neck with curvilinear probe in sagittal orientation and probe indicator directed cranially. The mentum of the mandible is indicated by the *solid arrow* and the hyoid bone is indicated by the *dashed arrow*. Deep to the hypoechoic mylohyoid and geniohyoid muscles (*single star*) lies the tongue (*double star*). The hyomental distance (HMD) is spanned by the *double-headed arrow*.

**Figure 3 diagnostics-13-01541-f003:**
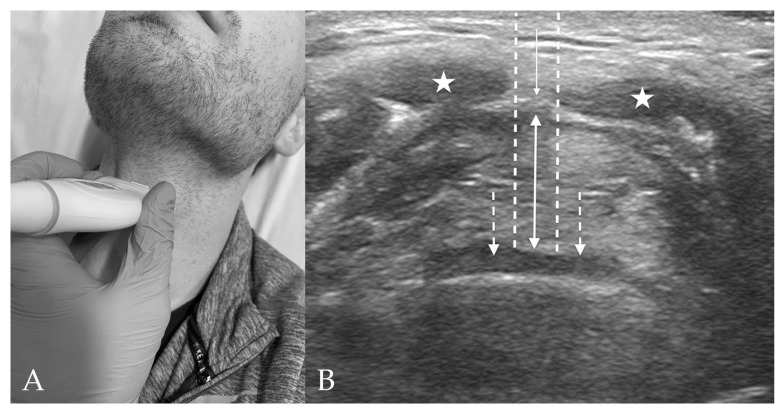
Thyrohyoid View. (**A**) Thyrohyoid probe placement on subject’s neck. (**B**) Thyrohyoid view of anterior neck with linear probe in transverse orientation. The pre-epiglottic space (*solid, double-headed arrow*) appears between the thyrohyoid membrane (*solid, single-headed arrow*) and the epiglottis (*dashed arrows*). The strap muscles (*stars*) are again visible superficially to the thyrohyoid membrane. The distance from skin to epiglottis (DSE) is spanned by the two *dashed lines*.

**Figure 4 diagnostics-13-01541-f004:**
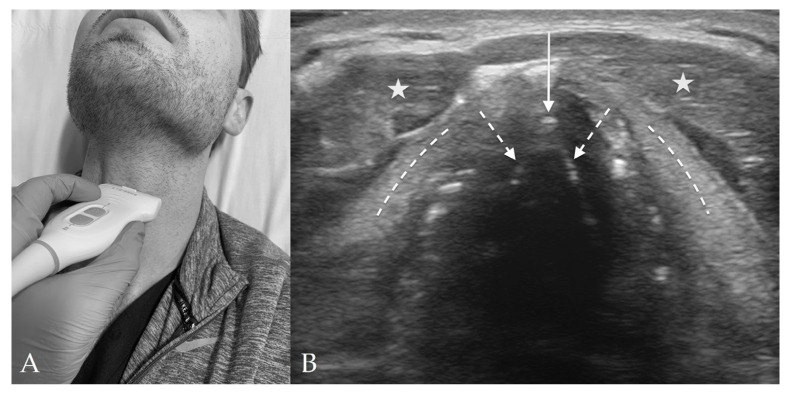
Thyroid View. (**A**) Thyroid probe placement on subject’s neck. (**B**) Thyroid view of anterior neck with linear probe in transverse orientation. The vocal cords (*dashed arrows*) join together at the anterior commissure (*solid arrow*). The thyroid cartilage (*dashed lines*) appears lateral to the vocal cords, and the strap muscles (*stars*) just superficial to the thyroid cartilage. Although not well visualized in this image, the arytenoids will usually appear at the posterior aspect of the bilateral vocal cords.

**Figure 5 diagnostics-13-01541-f005:**
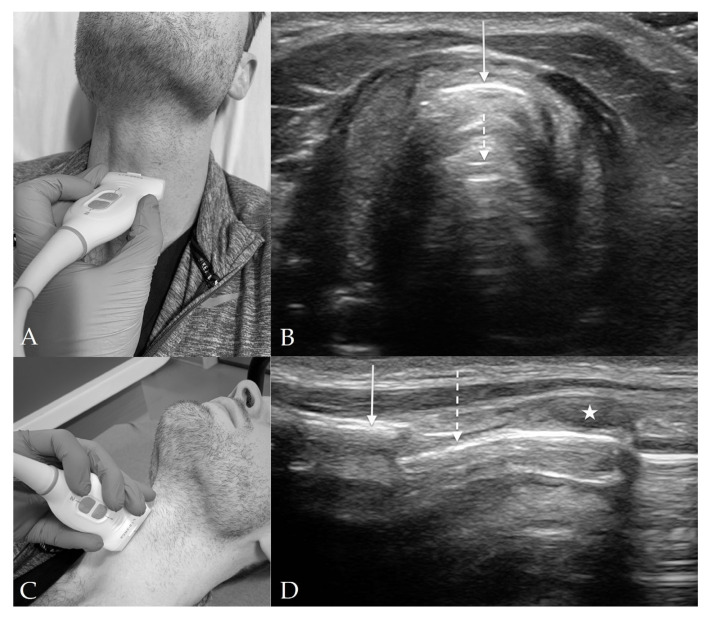
Cricothyroid View: Transverse and Sagittal Orientations. (**A**) Cricothyroid probe placement on subject’s neck in transverse orientation. (**B**) Cricothyroid view of anterior neck with linear probe in transverse orientation. The cricothyroid membrane (*solid arrow*) overlies the trachea with prominent reverberation artifact (*dashed arrow*) in the tracheal lumen. (**C**) Cricothyroid probe placement on subject’s neck in sagittal orientation. (**D**) Cricothyroid view of anterior neck with linear probe in sagittal orientation. The thyroid cartilage (*solid arrow*) appears superior to the hypoechoic cricoid cartilage (*star*). The cricothyroid membrane (*dashed arrow*) spans between the two cartilages.

**Figure 6 diagnostics-13-01541-f006:**
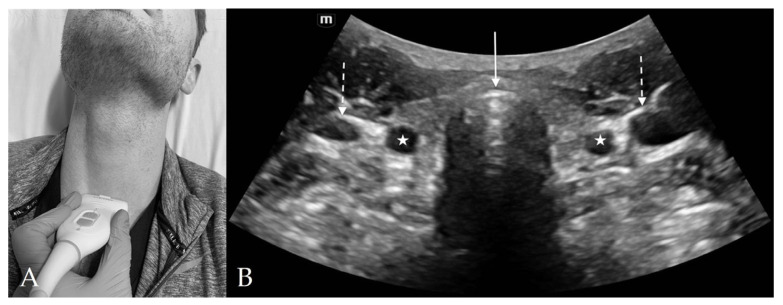
Suprasternal View. (**A**) Suprasternal probe placement on subject’s neck. (**B**) Suprasternal view of anterior neck with curvilinear probe. The tracheal cartilage (*solid arrow*) appears hyperechoic with reverberation artifact noted in the air-filled tracheal lumen posteriorly. The common carotid arteries (*stars*) and internal jugular veins (*dashed arrow*) appear laterally on each side of the trachea.

**Figure 7 diagnostics-13-01541-f007:**
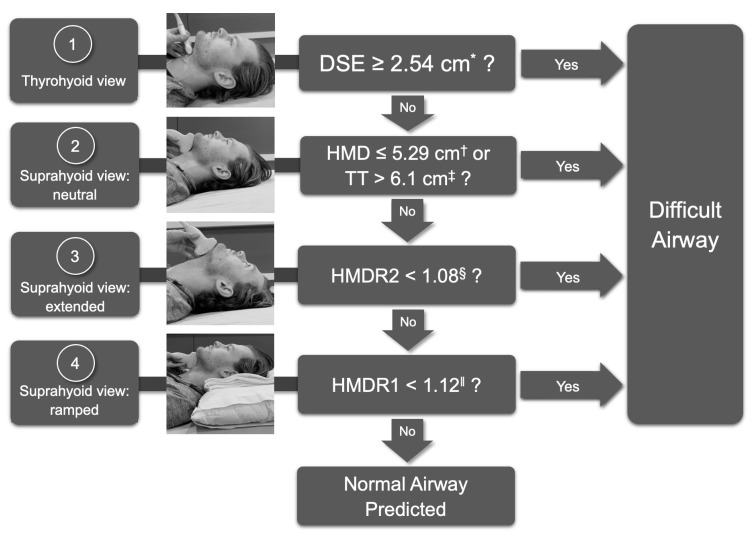
Difficult Airway Evaluation with Sonography (DARES) Protocol. Abbreviations: DSE, distance to epiglottis; HMD, hyomental distance; TT, tongue thickness; HMDR2, HMD extended/HMD neutral; HMDR1, HMD ramped/HMD neutral. * Falcetta, 2018 [16]; ^†^ Wu, 2022 [61]; ^‡^ Yao, 2017 [51]; ^§^ Rana, 2018 [44]; ^‖^ Petrisor, 2018 [60].

**Figure 8 diagnostics-13-01541-f008:**
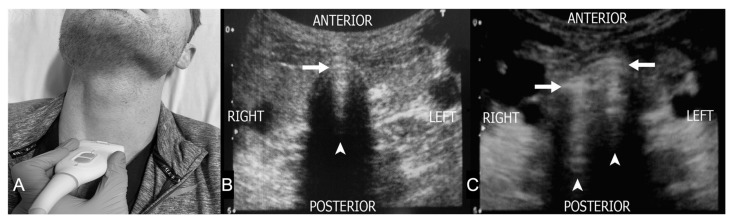
Confirmation of Endotracheal Intubation with Ultrasound. (**A**) Place the transducer transversely on the anterior neck at the level of the suprasternal notch for the best visualization and diagnostic accuracy. (**B**) Tracheal ETT position: only one A–M interface (*arrow*) with comet tail artifact (*arrowhead*) and posterior shadowing is observed (**C**) Esophageal ETT position: two A–M interfaces (*arrows*) with comet-tail artifacts (*arrowheads*) and posterior shadowing are noted (the “double tract” sign).

**Figure 9 diagnostics-13-01541-f009:**
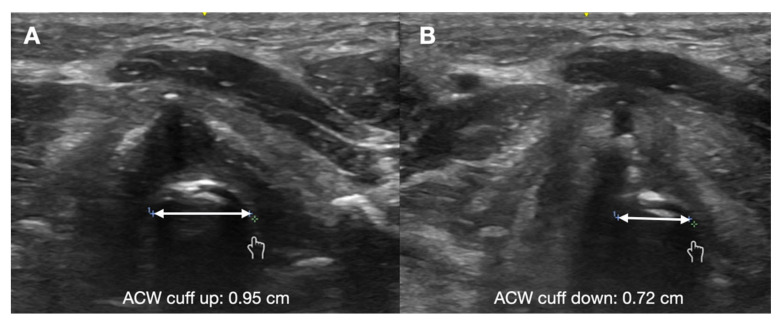
Air column width (*double-headed arrow*). (**A**) With ETT cuff up. (**B**) With ETT cuff down.

**Figure 10 diagnostics-13-01541-f010:**
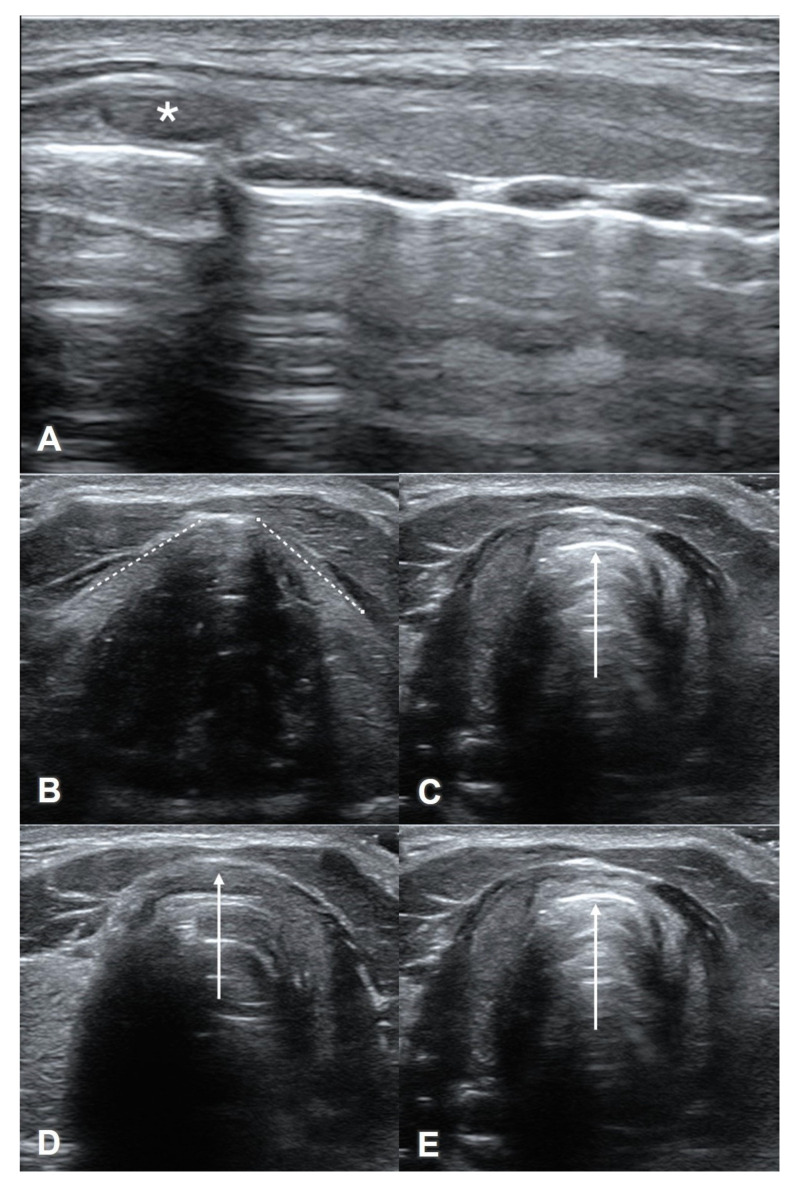
The longitudinal “string of pearls” technique and the transverse “thyroid-airline-cricoid-airline” (TACA) protocol to identify the CTM. (**A**) String of pearls with cricoid cartilage (*asterisk*). (**B**) “T” in “TACA” with outline thyroid cartilage (*dashed lines*). (**C**) “A” in “TACA” with hyperechoic CTM (*solid arrow*). (**D**) “C” in “TACA” with hypoechoic cricoid cartilage (*solid arrow*). (**E**) “A” in “TACA” with hyperechoic CTM (*solid arrow*).

**Table 1 diagnostics-13-01541-t001:** Summary of studies on ultrasonographic parameters for predicting difficult laryngoscopy or difficult intubation.

Author, Date, and Country	Setting	Patient Group	Study Design	Outcome Variable	Sample Size	Outcome Count	Key Results
Alessandri, 2019, Italy [29]	OR, elective surgery	Italian adults undergoing elective ENT surgery with general anesthesia	Prospective observational	CL grade; Han scale	194	Difficult airway = 34 (17.5% incidence) Difficult mask ventilation = 8 (4.1% incidence)	The best predictor of difficult airway was the minimum distance from the hyoid bone to skin surface
Altun, 2021, Turkey [31]	OR, elective surgery	Turkish adults undergoing an elective procedure with general anesthesia	Prospective observational	CL grade	140	Difficult laryngoscope = 22 (15.7% incidence)	Significant correlation of US measurements (ESD, HSD, and ToTR) to predict difficult airway, as well as significant correlation of US measurements (MP + HSD or MP + ESD) combined with IL to predict difficult airway (sensitivity 91%, specificity 97%)
Ambrose, 2022, India [32]	OR, elective surgery	Indian adults undergoing elective surgery with general anesthesia	Prospective, observational	CL grade	120	Difficult intubation = 41 (34.2% incidence)	Distance between skin to epiglottis, tongue thickness, and neck circumference were significantly associated with difficult intubation cases
Andruszkiewicz, 2016, Poland [33]	OR, elective surgery	Polish adults undergoing elective surgery	Prospective observational	CL grade	199	Difficult laryngoscope = 22 (11.1% incidence)	Strongest predictors were HMD in extension and tongue volume. Significant predictors were (1) HMD in neutral, (2) HMD in extension, (3) HMDR2, (4) tongue cross-sectional area, and (5) tongue volume
Agarwal, 2021, India [34]	OR, elective surgery	Indian adults undergoing	Prospective observational	Difficult intubation	1043	Difficult to intubate = 58 (5.56% incidence)	Skin to hyoid bone distance was most accurate to predict difficult intubation
Bindu, 2022, India [35]	OR, elective surgery	Indian adults with morbid obesity (BMI > 35) undergoing elective surgery with general anesthesia	Prospective observational	CL grade	70	Difficult laryngoscopy = 15 (21.4%)	Limited condylar mobility and increased TT were two independent ultrasonographic predictors for difficult direct laryngoscopy
Bouzid, 2022, Tunisia [36]	OR, elective surgery	Tunisian adults undergoing general anesthesia	Prospective observational	CL grade; Difficult intubation	200	Difficult laryngoscopy = 41 (incidence 20.5%), Difficult intubation = 29 (14.5%)	A combination of clinical criteria and ultrasound measurements increases the ability to predict difficult airway management
Chan, 2018, China [37]	OR, elective surgery	Chinese adults undergoing elective surgery with general anesthesia	Prospective observational	CL grade	113	Difficult laryngoscope = 39 (34.5% incidence)	Pre-E/aVF ratio (>1) had better accuracy than pre-E/mVF and pre-E/pVF. Pre-E/E-VC cutoff originally intended to study was too high for the population
Daggupati, 2022, India [38]	OR, elective surgery	South Indian adults undergoing elective operation with general anesthesia	Prospective observational	CL grade	310	Difficult laryngoscope = 62 (20.0% incidence)	Skin to epiglottis distance via US as part of an airway scoring system (using clinical predictors) was reliable in predicting DA (sensitivity 93%, specificity 85%)
Falcetta, 2018, Italy [16]	OR, elective surgery	Italian adults undergoing elective surgery with general anesthesia	Prospective observational	CL grade	301	Difficult laryngoscope = 28 (9.3% incidence)	US measured at >2.54 cm and PEA >5.04 cm^2^ may predict difficult laryngoscopes
Falsafi, 2023, Iran [39]	OR, elective surgery	Iranian adults undergoing elective surgery with general anesthesia	Prospective observational	CL grade; Mallampati score	120	Difficult laryngoscopy by CL = 34 (28.3% incidence) By Mallampati = 37 (30.8% incidence)	Neck circumference was significantly correlated with difficult laryngoscopy by CL. HMDN, HMDE, tongue thickness, OCH, and ST were significant by Mallampati
Lin, 2021, Taiwan [40]	OR, elective surgery	Taiwanese adults undergoing elective procedure under general anesthesia	Prospective observational	CL grade	47	Difficult laryngoscopy = 20 (42.6% incidence)	Submental ultrasound was not predictive in difficult laryngoscopy, only difficult mask ventilation (sensitivity 50%, specificity 87%)
Moura, 2021, Brazil [41]	OR, elective surgery (bariatric surgery)	Obese Adults	Prospective observational	CL grade	100	Difficult airway = 25 (25% incidence)	Skin to epiglottis distance (29.3 mm) was predictive for difficult intubation (AUC 0.656, sensitivity 66.7%, specificity 66%)
Ni, 2020, China [42]	OR, elective surgery	Chinese adults undergoing elective procedures with general anesthesia	Prospective observational	CL grade	211	Difficult laryngoscopy = 44 (20.9% incidence)	Significant predictor of difficult laryngoscopies using DSE (sensitivity 81.8%, specificity 85.6%)
Prathep, 2022, Thailand [43]	OR, elective surgery	Obese adults in Thailand	Prospective observational	CL grade	88	Difficult laryngoscopy = 13 (14.8% incidence)	Scoring model based on US measurements were predictive of difficult laryngoscopy (AUC 0.77)
Rana, 2018, India [44]	OR, elective surgery	Indian adults undergoing elective surgery with general anesthesia	Prospective observational	CL grade	120	Difficult airway = 28 (23.3% incidence)	Significant predictors were Pre-E/E-VC ratio and HDMR2. Pre-E/E-VC was stronger
Reddy, 2016, India [45]	OR, elective surgery	Indian adults undergoing elective surgery with general anesthesia	Prospective observational	CL grade	100	Difficult laryngoscope = 14 (14% incidence)	Significant predictors were skin-to-anterior commissure of VC and Pre-E/E-VC ratio
Wang, 2019, China [46]	OR, elective surgery	Chinese adults undergoing elective procedures with general anesthesia	Prospective observational	CL grade	499	Difficult laryngoscopy = 47 (9.4% incidence)	Angle between the epiglottis and glottis (less than 50 degrees) had the best sensitivity (81%) and specificity (89%)
Wang, 2022, China [47]	OR, elective surgery	Chinese adults undergoing elective procedure under general anesthesia	Prospective observational	Cl grade	2254	Difficult laryngoscope = 142 (6.3% incidence) Difficult intubation = 51 (2.3% incidence)	Study nomogram (consisting of US measurements and clinical parameters) had AUC 0.933 for difficult laryngoscopy and 0.974 for difficult intubation
Wu, 2014, China [48]	OR, elective surgery	Han Chinese adults aged 20–65, scheduled to undergo general anesthesia	Prospective observational	CL grade	203	Difficult laryngoscope = 28 (13.8% incidence)	Independent predictors of difficult airway were the distance from skin to (1) hyoid bone, (2) epiglottis, and (3) anterior commissure of VC
Wu, 2022, Taiwan [49]	OR, elective surgery	Taiwanese adults undergoing laparoscopic sleeve gastrectomy	Prospective observational	CL grade	80	Difficult laryngoscopy = 17 (21% incidence)	Greater neck circumference was independently associated with difficult laryngoscopy in obese patients. CSA of the tongue base may serve as an index to identify high-risk patients before tracheal intubation
Yadav, 2019, India [50]	OR, elective surgery	Indian adults undergoing elective procedures under general anesthesia	Prospective observational	CL grade	310	Difficult laryngoscopy = 35 (11.3% incidence)	Tongue thickness as a predictor (sensitivity 69.6%, specificity 77%); skin to hyoid bone distance (sensitivity 68%, specificity 73%)
Yao, 2017, China [51]	OR, elective surgery	Han Chinese adults undergoing elective surgery with general anesthesia	Prospective observational	Difficult intubation; CL grade	2254	Difficult laryngoscope = 142 (6.3% incidence) Difficult intubation = 51 (2.26% incidence)	Tongue thickness > 6.1 cm is an independent predictor for difficult laryngoscopy (sensitivity 0.75, specificity 0.72). Significant predictors were tongue thickness and tongue thickness/TMD ratio
Yao, 2017, China [52]	OR, elective surgery	Han Chinese adults undergoing elective surgery with general anesthesia	Prospective observational	CL grade	484	Difficult laryngoscope = 41 (8.5% incidence) Difficult intubation = 5 (1.0% incidence)	Condylar translation was a significant predictor of difficult laryngoscopy. Condylar translation at ≤1 cm is a meaningful TMJ mobility evaluation predicting difficult laryngoscopy
Zheng, 2021, China [53]	OR, elective surgery	Han Chinese adults undergoing elective procedures with general anesthesia	Prospective observational	CL grade	230	Difficult laryngoscopy = 28 (12.2% incidence) Difficult intubation = 12 (5.2% incidence)	Midsagittal tongue CSA had highest predictor value for difficult laryngoscopy (sensitivity 71%, specificity 60%) and difficult intubation (sensitivity 39%, specificity 89%)

Abbreviations: AUC, area under the curve; BMI, body mass index; CL, Cormack-Lehane; CSA, cross-sectional area; DA, difficult airway; DSE, distance from skin to epiglottis; ENT, ear, nose, and throat (otolaryngology); ESD, median distance from skin to epiglottis; HMD, hyomental distance; HMDE, hyomental distance with the head in the extended position; HMDN, hyomental distance with the head in the neutral position; HMDR2, hyomental distance in the extended position to that in the neutral position ratio; HSD, hyoid bone skin distance; OCH, oral cavity height; OR, operating room; PEA, pre-epiglottic area; Pre-E, pre-epiglottic distance; Pre-E/aVF, distance from the epiglottis to the anterior vocal folds; Pre-E/E-VC, ratio of pre-epiglottis space distance (Pre-E) and the distance between the epiglottis and the vocal folds; pre-E/pVC, ratio of pre-epiglottis space to distance between epiglottis and posterior vocal folds; ST, distance from skin to thyroid cartilage; THM, thyrohyoid membrane; TMD, thyromental distance; TMJ, temporomandibular joint; ToTR, thickness of tongue root; TT, tongue thickness; US, ultrasound; VC, vocal cords.

**Table 2 diagnostics-13-01541-t002:** Summary of studies on the use of ultrasound to confirm endotracheal tube placement.

Author, Date, and Country	Setting	Patient Population	Study Type	Sample Size	Outcomes	Key Results
Abdelrahman, 2020, Egypt [68]	OR	Adult	Prospective, Randomized	70	Ultrasound and Capnography	93.8% sensitive, 66.7% specific, 91.4% accurate
Afzalimoghaddam, 2019, Iran [69]	ED	Adult	Prospective, observational	90	Ultrasound and Capnography	100% sensitive, 100% specific, US is easily taught and learned, US is an effective method for confirming ET tube placement in ED and does not require interruption of chest compressions
Arafa, 2018, Egypt [70]	OR	Adult	Prospective observational	107	Ultrasound, Capnography, and Auscultation	97% sensitive, 71.4% specific
Arya, 2018, USA [71]	ICU	Adult	Prospective observational	75	Ultrasound and Capnography	83% sensitive, 100% specific, 100% PPV, 97% NPV
Chen, 2018, China [72]	OR	Adult	Prospective case-control	105	Ultrasound and Direct laryngoscopy	100% sensitive, 100% specific, 100% accurate, allows proper positioning of ETT cuff and reduces vocal cord injury after intubation
Chen, 2020, China [73]	ICU	Adult	Prospective observational	118	Ultrasound and Fiberoptic Bronchoscopy	75% sensitive, 100% specific, 100% PPV, 97.2% NPV, allowed for rapid clinical decision making
Chou, 2011, Taiwan [9]	ED	Adult	Prospective observational	112	Ultrasound and Capnography	98.2% accuracy, 98.9% sensitive, 94.1% specific
Chou, 2013, Taiwan [74]	ED	Adult (cardiac arrest)	Prospective observational	89	Ultrasound, Capnography, and Auscultation	100% sensitive, 85.7% specific
Chowdhury, 2020, India [75]	OR	Adult	Prospective, single blinded clinical trial	120	Ultrasound, Capnography, and Auscultation	100% sensitive, 100% specific, US was fastest method of determining ET intubation, allowed for rapid correction of errors in placement
Inangil, 2018, Turkey [76]	OR	Adult	Prospective, paired, single blind	56	Ultrasound and Capnography	95.75% sensitive, 100% specific, faster than capnography
Kabil, 2018, Saudi [77]	ICU	Adult	Prospective observational	40	Ultrasound and Bronchoscopy	97.2% sensitive, 100% specific, rapidly informed decision making
Kad, 2018, India [78]	OR	Adult	Prospective crossover	100	Ultrasound and Auscultation	88.46% sensitive, 100% specific when “lung sliding” sign is visualized on US, superior to auscultation alone
Lahham, 2017, USA [79]	ED	Adult	Prospective cohort	72	Ultrasound and Capnography	98.5% Sensitive, 75% specific
Masoumi, 2017, Iran [80]	ED	Adult	Prospective observational	100	Ultrasound and Capnography	98.9% sensitive, 100% specific for US (TRUE method) to detect ETT
Mousavi, 2022, Iran [81]	ED	Adult	Cross-sectional	66	Ultrasound and Capnography	98% specific, 66% sensitive, suprasternal view better than subxiphoid for placement confirmation
Patil, 2019, India [82]	ICU	Adult	Prospective observational	89	Ultrasound and Capnography	96% sensitive, 100% specific; allowed for real time detection
Rahmani, 2017, Iran [83]	ED	Adult	Descriptive analytic	75	Ultrasound and Capnography	“Snowstorm” sign was 100% sensitive and 100% specific for detecting ETT
Rahul, 2016, USA [84]	ICU	Adult	Prospective, double blind	20	Ultrasound, Capnography, and Auscultation	100% accuracy, Outperformed gold standard, able to detect right main stem intubation
Sethi, 2019, India [85]	OR	Adult	Prospective, randomized, observational	90	Ultrasound, Capnography, and Auscultation	100% accuracy, faster than capnography
Thomas, 2017, India [86]	ED	Adult	Prospective cohort study	100	Ultrasound and Capnography	97.89% sensitive, 100% specific for diagnosis of esophageal intubation, equal to capnography but quicker
Yang, 2017, China [87]	OR	Adult	Prospective, randomized, double blinded	93	Ultrasound and Capnography	100% sensitive, 88.9% specific for US localization of tracheal intubations
Zamani, 2017, Iran [88]	ED	Adult	Prospective, cross sectional	150	Ultrasound, Auscultation, Direct Laryngoscopy, ETT aspiration, and Pulse Oximetry	96% sensitive, 88% specific, 98% PPV, 78% NPV for tracheal US in ETT confirmation
Zamani, 2018, Iran [89]	ED	Adult trauma	Prospective, single blind	100	Ultrasound and Capnography	97.9% sensitive, 83.3% specific

Abbreviations: ED, emergency department; ETT, endotracheal tube; ICU, intensive care unit; NPV, negative predictive value; OR, operating room; PPV, positive predictive value; TRUE, tracheal rapid ultrasound exam; US, ultrasound.

## Data Availability

Not applicable.

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
