# Peer review of "Point-of-Care Ultrasound in Airway Evaluation and Management: A Comprehensive Review"

_diagnostics, 2023, doi:10.3390/diagnostics13091541_

Round 1

Reviewer 1 Report

The main question addressed by the research is use of POCUS in airway management; prediction of difficulty, as well as aid in airway management.

I consider the topic VERY relevant, it does address the gap in the field. Or at least solidifies review of evidence and appropriate guidance.

It adds excellent EBM review and gives specific guidance on actual performance of POCUS.

Very good and comprehensive review. Nothing to add. Accepted. Congratulations to authors!

Author Response

Thanks for your comments.

Reviewer 2 Report

Dear Editor-in-Chief

Judy Lin et al. present in their review “Point-of-Care Ultrasound in Airway Evaluation and Management: A Comprehensive Review” a comprehensive and state of the art review about the sonographic evaluation and management of the airway. I am a big fan and supporter of ultrasound myself, but some questions are unanswered by the authors.

First, are there any limitations of the sonographic evaluation of the airway? The “models” in the figures are all thin, but the common ICU patient is not. Are all these steps also possible in obese ICU patients? Are there any other limitations to the sonographic approach? Trauma patients?

Second, how much education is needed until one can safely perform your proposed protocol? Did you check that before you implanted this protocol at your ED? You speak of 1mm differences (2.54cm). I guess you must be very skilled to do that?

Third, the title of the review includes the word “management”. I am missing a passage about the management of a difficult airway or at least mention Difficult-Airway-Society (DAS) guidelines. You should mention that ultrasound is a very safe and useful tool, but physicians must be trained in bag mask ventilation, larynx masks and video-laryngoscopy as well! Patients don`t die from failed intubation but from failed oxygenation.

Nonetheless, the review is well written. I would recommend it for publication after the mentioned points.

Minor Flaws

I have only found a few typos:

Line 6: Eric H Chou and MD à is there an author missing?

Line 287: DSE[61]. à gap

Line 326: death.[62] à reference before the end of the sentence.

Line 414: outer diameter <1 à mm? cm?

Figures:

Please use one graphical approach for all figures.

Abstract:

After reading the abstract I wasn`t sure if this review addresses airway evaluation in the emergency department or/and in the intensive care. Patients are not the same in the ED or in the ICU which might lead to a different approach in the ICU or in the ED when it comes to airway evaluation.

Author Response

Dear Editor and Reviewers,

Thank you for your comments. We have revised the manuscript as advised. The detailed point-by-point responses are as follows. We hope the manuscript is now acceptable to Diagnostics.

Judy Lin et al. present in their review “Point-of-Care Ultrasound in Airway Evaluation and Management: A Comprehensive Review” a comprehensive and state of the art review about the sonographic evaluation and management of the airway. I am a big fan and supporter of ultrasound myself, but some questions are unanswered by the authors. 

First, are there any limitations of the sonographic evaluation of the airway? The “models” in the figures are all thin, but the common ICU patient is not. Are all these steps also possible in obese ICU patients? Are there any other limitations to the sonographic approach? Trauma patients?

Reply: Thank you for your question. The use of ultrasound is important for airway assessment in obese patient population due to the difficulty in identifying landmarks using traditional physical exam.1,2 Prior studies have shown that ultrasound can accurately predict difficult laryngoscopy among obese patients.3-5 Bindu et al. demonstrated that limited condylar mobility and increased tongue thickness are independent sonographic predictors of difficult direct laryngoscopy in morbidly obese patients.5 Moura et al. showed that using ultrasound method as a predictor of difficult intubation is promising in patients with obesity.4 In addition, previous studies showed that emergency physicians were able to rapidly and reliably mark the cricothyroid membrane in obese patients using ultrasound.6,7 Thus, we believe that these ultrasound scanning steps are possible and essential in obese ICU patients. In multiple trauma patients, although ultrasound can achieve high sensitivity and specificity to confirm ETT placement, the application can be limited in patients with neck trauma with subcutaneous emphysema, laryngeal injury, or foreign bodies.8 We have described the limitations and revised the manuscript.

Second, how much education is needed until one can safely perform your proposed protocol? Did you check that before you implanted this protocol at your ED? You speak of 1mm differences (2.54cm). I guess you must be very skilled to do that?

Reply: Thank you for your question. To date, there are limited literatures to define the learning curve of airway ultrasound in difficult airway assessment. Gottlieb et al. demonstrated that ultrasound can be accurate for airway assessment if performed by an ultrasound expert. Among novices, the ultrasound technique can be accurate in thin, but appears less accurate in obese cadavers.9 Prior studies showed that emergency medicine resident physicians who received a 1-hour standard airway ultrasound lecture and 8-hour hands-on practice can successfully use ultrasound for airway assessment.10,11 Since ultrasound scanning can be highly operator dependent, we believe a specific airway ultrasound lecture with hands-on practice is required before applying airway ultrasound in clinical practice. We have revised the manuscript.

Third, the title of the review includes the word “management”. I am missing a passage about the management of a difficult airway or at least mention Difficult-Airway-Society (DAS) guidelines. You should mention that ultrasound is a very safe and useful tool, but physicians must be trained in bag mask ventilation, larynx masks and video-laryngoscopy as well! Patients don`t die from failed intubation but from failed oxygenation.

Reply: Thank you for your comment. We have revised the manuscript accordingly.

Nonetheless, the review is well written. I would recommend it for publication after the mentioned points.

Minor Flaws

I have only found a few typos: 

Line 6: Eric H Chou and MD à is there an author missing?

Reply: Thank you for your question. We have revised the manuscript.

Line 287: DSE[61]. à gap

Reply: Thank you for your comment. We have revised the manuscript accordingly.

Line 326: death.[62] à reference before the end of the sentence.

Reply: Thank you for your comment. We have revised the manuscript accordingly.

Line 414: outer diameter <1 à mm? cm?

Reply: Thank you for your question, we have revised the manuscript.

“Shinohara, et al. found that the glottis to the ETT outer diameter ratio <1 in females performed within three hours prior to or after extubation in 212 patients was associated with airway obstruction symptoms”

Figures:

Please use one graphical approach for all figures.

Reply: Thank you for your comment. We have revised the Figures accordingly.

Abstract:

After reading the abstract I wasn`t sure if this review addresses airway evaluation in the emergency department or/and in the intensive care. Patients are not the same in the ED or in the ICU which might lead to a different approach in the ICU or in the ED when it comes to airway evaluation.

Reply: Thank you for your comment. We have revised the abstract accordingly.

“Airway management is a common and critical procedure in the acute setting such as in the Emergency Department (ED) or Intensive Care Unit (ICU).”

Reference:

  1. Falcetta S, Cavallo S, Gabbanelli V, et al. Evaluation of two neck ultrasound measurements as predictors of difficult direct laryngoscopy: A prospective observational study. European journal of anaesthesiology. 2018;35(8):605-612.
  2. Carsetti A, Sorbello M, Adrario E, Donati A, Falcetta S. Airway Ultrasound as Predictor of Difficult Direct Laryngoscopy: A Systematic Review and Meta-analysis. Anesthesia and analgesia. 2022;134(4):740-750.
  3. Prathep S, Jitpakdee W, Woraathasin W, Oofuvong M. Predicting difficult laryngoscopy in morbidly obese Thai patients by ultrasound measurement of distance from skin to epiglottis: a prospective observational study. BMC Anesthesiol. 2022;22(1):145.
  4. Moura ECR, Filho ASM, de Oliveira E, et al. Comparative Study of Clinical and Ultrasound Parameters for Defining a Difficult Airway in Patients with Obesity. Obes Surg. 2021;31(9):4118-4124.
  5. Bindu HM, Dogra N, Makkar JK, Bhatia N, Meena S, Gupta R. Limited condylar mobility by ultrasonography predicts difficult direct laryngoscopy in morbidly obese patients: An observational study. Indian J Anaesth. 2021;65(8):612-618.
  6. Kristensen MS, Teoh WH, Rudolph SS, Hesselfeldt R, Børglum J, Tvede MF. A randomised cross-over comparison of the transverse and longitudinal techniques for ultrasound-guided identification of the cricothyroid membrane in morbidly obese subjects. Anaesthesia. 2016;71(6):675-683.
  7. Nicholls SE, Sweeney TW, Ferre RM, Strout TD. Bedside sonography by emergency physicians for the rapid identification of landmarks relevant to cricothyrotomy. The American journal of emergency medicine. 2008;26(8):852-856.
  8. Zamani M, Esfahani MN, Joumaa I, Heydari F. Accuracy of Real-time Intratracheal Bedside Ultrasonography and Waveform Capnography for Confirmation of Intubation in Multiple Trauma Patients. Adv Biomed Res. 2018;7:95.
  9. Gottlieb M, Bailitz JM, Christian E, et al. Accuracy of a novel ultrasound technique for confirmation of endotracheal intubation by expert and novice emergency physicians. The western journal of emergency medicine. 2014;15(7):834-839.
  10. Chou HC, Chong KM, Sim SS, et al. Real-time tracheal ultrasonography for confirmation of endotracheal tube placement during cardiopulmonary resuscitation. Resuscitation. 2013;84(12):1708-1712.
  11. Chou HC, Tseng WP, Wang CH, et al. Tracheal rapid ultrasound exam (T.R.U.E.) for confirming endotracheal tube placement during emergency intubation. Resuscitation. 2011;82(10):1279-1284.